# Gelatin Improves the Performance of Oregano Essential Oil Nanoparticle Composite Films—Application to the Preservation of Mullet

**DOI:** 10.3390/foods12132542

**Published:** 2023-06-29

**Authors:** Yuan Ma, Siqi Chen, Ping Liu, Yezheng He, Fang Chen, Yifan Cai, Xianqin Yang

**Affiliations:** Sichuan Key Laboratory of Food Biotechnology, School of Food and Bioengineering, Xihua University, Chengdu 610039, China; 18328373983@163.com (S.C.); liuping20210414@163.com (P.L.); heyezheng666@163.com (Y.H.); cf2171819250@163.com (F.C.); 18180580357@163.com (Y.C.); 19946801453@163.com (X.Y.)

**Keywords:** oregano essential oil nanoparticles, gelatin, mullet preservation

## Abstract

In this study, the addition of oregano oil chitosan nanoparticles (OEO-CSNPs) was conducted to enhance the comprehensive properties of gelatin films (GA), and the optimal addition ratio of nanoparticles was determined for its application in the preservation of mullet. Oregano oil chitosan nanoparticles were organically combined with gelatin at different concentrations (0%, 2%, 4%, 6% and 8%) to obtain oregano oil–chitosan nanoparticle–GA-based composite films (G/OEO-CSNPs), and thereafter G/OEO-CSNPs were characterized and investigated for their preservative effects on mullet. Subsequent analysis revealed that OEO-CSNPs were uniformly dispersed in the GA matrix, and that G/OEO-CSNPs had significantly improved mechanical ability, UV-visible light blocking performance and thermal stability. Furthermore, the nanoparticles exhibited excellent antioxidant and antibacterial properties, and they improved the films’ suitability as edible packaging. The attributes of the G/OEO-CSNPs were optimized, the films had the strongest radical scavenging and lowest water solubility, and electron microscopy also showed nanoparticle penetration into the polymer when the concentration of OEO-CSNPs was 6% (thickness = 0.092 ± 0.001, TS = 47.62 ± 0.37, E = 4.06 ± 0.17, water solubility = 48.00 ± 1.11). Furthermore, the GA-based composite film containing 6% OEO-CSNPs was able to inhibit microbial growth, slow fat decomposition and protein oxidation, reduce endogenous enzyme activity, and delay the spoilage of mullet during the refrigeration process, all of which indicate its excellent potential for meat preservation application.

## 1. Introduction

Mullet is a common and high-quality freshwater fish. It is rich in protein, polyunsaturated fatty acids and various amino acids [1]. However, the high water and protein content in the harvested fish enables a significant degree of enzymatic activity and consequent deterioration, leading to serious economic losses [2]. The development of novel edible coatings made of natural macromolecules, such as polysaccharides, fats and proteins, has become an important topic in food preservation studies [3,4]. Protein-based film materials have been shown to have a higher nutritional value and better film formation capacity than other potential materials [5] and are, therefore, considered one of the most promising biodegradable bio-based options for food packaging [6]. However, the thermal, mechanical and water barrier properties of GA films are poor [7], providing insufficient mechanical strength and barrier protection for food products [8], and their hygroscopic properties limit their application in high-moisture foods [9].

The use of essential oils in edible coatings, with their active polymer systems forming nanoparticles that can deliver essential bioactive compounds, has been shown to reduce oxidation, water changes and the loss of food texture or aroma induced by molecular activities such as oxidation, carbon dioxide and water vapor [10], while also controlling the release of antimicrobial agents. Such edible coatings, thus, continue to function as active substances during food storage. Chitosan nanoparticles containing natural plant essential oils can improve the mechanical and barrier properties of GA films [11,12]. Furthermore, the combination of bio-composite films with nanomaterials can significantly improve the strength of biofilms [13].

As of yet, however, no research has reported an improvement in the comprehensive properties of GA-based composite membranes through the addition of oregano essential oil chitosan nanoparticles (OEO-CSNPs). Therefore, in this study, OEO-CSNPS were added to GA to improve its film properties, and subsequently applied in the preservation of harvested mullet. The effects of the incorporation of OEO-CSNPs onto composite films (G/OEO-CSNPs) were evaluated to ascertain and determine the consequential effect of GA-based composite films with the appropriate OEO-CSNP contents on the sensory, microbiological and physicochemical qualities of fresh mullet meat during storage at 4 °C. The aims of this study were to provide environmentally sound, healthy and safe products for consumers, to provide a theoretical breakthrough in the development of food preservation films, and to provide new ideas for quality control.

## 2. Materials and Methods

### 2.1. Materials

Gelatin (jelly power 250 g, food grade) was purchased from Shangshui County Fuyuan Gelatin Co. (Zhoukou, China); OEO was purchased from Ji’an Huashen Fragrance Oil Co., Ltd. (Ji’an, China); chitosan (CS) (90% deacetylated) was provided by Shanghai Yuanye Biotechnology Co., Ltd. (Shanghai, China); sodium tripolyphosphate (TPP) was purchased from Shandong Yousu Chemical Technology Co., Ltd. (Heze, China); and nutrient agar was purchased from Beijing Auboxing Biotechnology Co. (Beijing, China). Unless otherwise stated, all chemicals and solvents were of analytical grade and were purchased from Chengdu Kolon Chemical Co., Ltd. (Chengdu, China) The mullet (2000 g) was purchased from the market in Pidu District, Chengdu, China. The fish were decapitated, gutted and skinned, and the fish muscles were immediately used for the experiments in this study.

### 2.2. Preparation of OEO-CSNPs

The nanoparticles were prepared via ionic cross-linking, with reference to the method described by Ma et al. [10].

### 2.3. Preparation of Bioactive Films

The GA films were prepared according to the method described by Hosseini et al. [14] with slight modifications. First, 3 g of GA was dissolved in 100 mL of distilled water for 1 h. The GA solution was prepared by heating with continuous stirring at 45 °C for 30 min, whereafter we added 30% glycerin (*w*/*w* of gelatin) as a plasticizer. The solution was heated again and stirred at 45 °C for 15 min. Then, different levels of GA-based OEO-CSNPs (0, 2, 4, 6 and 8%, *w*/*w*) were dispersed into distilled water (50 mL) and sonicated for 15 min to obtain various ratios of OEO-CSNP suspensions. The suspension was then added dropwise to the GA solution (50 mL) and gently stirred for 60 min, whereafter the film-forming dispersion was degassed under vacuum conditions for 15 min to remove any air bubbles. Finally, 20 mL of the film-forming dispersion was poured into Petri dishes (9 cm × 9 cm) and dried at room temperature (23–25 °C) for three days to obtain the G/OEO-CSNP membranes. The membranes were then removed, stored at a specific temperature and humidity (25 °C, 40% RH) for 48 h and, finally placed in a desiccator until required for analysis.

### 2.4. Scanning Electron Microscopy (SEM)

The microscopic morphologies of the coated film surfaces and fractured surfaces were observed via a field emission scanning electron microscope (Gemini SEM 300, ZEISS, Jena, Germany) at an accelerating voltage of 3.00 kV. The film samples were lysed in liquid nitrogen, the fracture surfaces of the films were coated well with gold spray for 3 min, and the films were then fixed onto the sample bench for detection.

### 2.5. X-ray Diffraction (XRD)

Analysis was performed using an X-ray diffractometer (D8 Advance, Bruker, Germany). The films were scanned at 40 kV and 30 mA at a scan rate of 2°/min in the range of (2θ) = 5–50°.

### 2.6. Fourier Transform Infrared Spectroscopy (FTIR)

The films were analyzed according to 32 consecutive FTIR spectra in the wavelength range from 4000 cm^−1^ to 400 cm^−1^ with a resolution of 4 cm^−1^ using an FTIR spectrometer (Nicolet AVATAR 380, Thermo Fisher Scientific, Waltham, MA, USA) [15].

### 2.7. Mechanical Properties

The treated samples (1.5 cm × 8 cm) were placed onto an Instron 5944 electronic universal testing machine (Instron, MA, USA; clamping distance 40 mm and measuring speed 10 mm/min) to determine the tensile strength (TS or strength of extension) and elongation at break (EB) for the films. The experiments were repeated five times, after which the average values were calculated. TS was calculated using Equation (1):(1)$$TS(MPa)=\frac{Fm}{d\times b}\times 100$$
where TS is the tensile strength (KPa), *F_m_* is the maximum tensile force at break (N), *b* is the width of the film sample (mm), and *d* is the thickness of the film sample (mm).

Elongation (E) was calculated according to Equation (2):(2)$$E\;(\%)\;=\frac{L_{1}-L_{0}}{L_{0}}\times 100$$
where E is the elongation at break (%), *L*_0_ is the length of the sample before stretching (m), and *L*_1_ is the maximum length reached when the film breaks (mm).

### 2.8. Thickness and Moisture Content

The thickness of the film samples was measured with the use of vernier calipers, and the average measurements of at least five random positions at each sample were taken as the average film thickness.

Referring to the method described by Riaz et al. [16], the film sheets (3 cm × 3 cm) were dried to a constant weight at 105 °C and recorded as *M*_1_. They were then placed in a beaker containing 50 mL distilled water and left to stand at room temperature for 24 h, after which they were removed, dried to a constant weight at 105 °C and recorded as *M*_2_. Film solubility was calculated as shown in Equation (3):(3)$$\mathrm{Film}\;\mathrm{solubility}\;(\%)\;=\frac{M_{1}-M_{2}}{M_{1}}\times 100$$

### 2.9. Ultraviolet (UV)-Visible Absorbance and Transmittance

The film samples were cut into strips (3 cm × 1 cm) and their UV-visible absorbance and transmittance were measured in the wavelength range of 200–800 nm using an UV-visible spectrophotometer (UV-2400, Shimadzu, Kyoto, Japan). This was performed according to the method described by Tavassoli et al. [17]. 

### 2.10. Thermogravimetric Analysis (TGA) Test

The thermal stability values of 3–5 mg samples were analyzed using a thermogravimetric analyzer (DTG-60, Shimadzu, Japan). The nitrogen gas flow rate was 30 mL/min at a heating rate of 10 °C/min from 30 °C to 600 °C.

### 2.11. Antioxidant Properties

The antioxidant activity of the film was analyzed via both DPPH and ABTS spectrophotometric assays. For the scavenging of DPPH radicals, 20 mg of membrane samples were mixed with 4 mL of DPPH methanol solution (25 mg/L). Incubation was performed at room temperature for 1 h and absorbance was measured at 517 nm [16].

In the ABTS analysis, potassium persulfate was added to a 7 mM ABTS solution to produce a final concentration of 2.45 mM. The solution was then incubated for 16 h at room temperature in the dark. The absorbance of the stable ABTS radical storage solution was adjusted to 0.70 ± 0.02 using an acetate buffer solution before measurement, after which 20 mg of membrane sample was added to 4 mL of ABTS solution. This solution was then incubated for 30 min at room temperature, and the absorbance was measured at 734 nm [18]. A film-free solution was used as the control test. In both assays, free radical scavenging activity (RSA) was calculated using Equation (4):(4)$$\mathrm{RSA}\;(\%)\;=\frac{A_{c}-A_{s}}{A_{c}}\times 100$$
where *A_c_* is the absorbance of the blank and *A_s_* is the absorbance of the sample.

### 2.12. Antimicrobial Activity Test

In the preliminary experiments, we obtained four strains of dominant spoilage bacteria through the isolation, screening and identification of the dominant spoilage bacteria, the observation of colony morphology, performing physiological and biochemical experiments and 16 S rDNA sequence analysis.

The inhibition of the growth of dominant spoilage bacteria by the GA/OEO-CSNP composite membranes was investigated according to the procedure described in a previous study [19]. First, the four dominant bacteria were inoculated in 20 mL of sterile tryptic soy broth (TSB). The solution was then incubated at 30 °C for 21 h, and centrifuged at 5000 rpm for 10 min. Thereafter, the cell precipitates were removed and repositioned in 100 mL of sterile TSB broth and diluted with sterile distilled water to the bacterial concentration (10^6^–10^7^ CFU/mL). Finally, 50 mL of the diluted broth was placed in a conical flask, 100 mg of membrane sample was added, and the solution was incubated with gentle shaking at 30 °C for 12 h. The antibacterial performance of the membranes was tested by counting colonies on the plates at 3 h intervals, using the broth without a membrane sample but at the same dilution as the control.

### 2.13. Preservation of Mullet

#### 2.13.1. Preparation of Fish Samples

The fish meat was cleaned with sterile water and cut into 3 × 3 × 1 cm^3^-sized samples under aseptic conditions, after which the cut pieces were stored at 4 °C for 10 min and then drained. The slices were then randomly divided into the following four groups: CK (untreated fish); GA (fish treated with GA film wrapping without the addition of nanoparticles); OEO-CSNPs (fish samples treated with the same concentration of nanoparticle coating); and G/OEO-CSNPs (fish treated with nanoparticle GA film wrapping and then placed individually in disposable Petri dishes). All fish samples were stored at 4 °C for 12 days and analyzed on days 0, 3, 6, 9 and 12 [20].

#### 2.13.2. Sensory Evaluation

All samples were assessed by 10 specially trained personnel according to the following sensory attributes and grades: elasticity (1, inelastic to 10, elastic); muscle morphology (1, not dense/loose to 10, dense/clear texture); odor (1, strong fishy odor to 10, freshness/fresh taste); and color (1, dull to 10, bright and shiny).

#### 2.13.3. pH Measurement

First, 10 g mullet samples were weighed in a clean, dry beaker, to which 10 times the weight of the sample in potassium chloride solution (KCl) (0.1 mol/L) was added. The mixture was vortexed for 1 min and then left for 30 min. The pH value of the supernatant was then measured using a pH meter (PHS-320, Chengdu Century Ark Technology Co., Chengdu, China) at 25 ± 2 °C.

#### 2.13.4. Determination of Thiobarbituric Acid Reactive Substances (TBARS)

TBARS were determined with reference to the method of Wu et al. [21] with slight modifications. Briefly, 5 g of filleted fish samples was chopped with a knife, mixed with 20 mL of 7.5% (*w*/*v*) trichloroacetic acid (TCA) and then centrifuged at 5000 rpm for 1 min at 4 °C. This mixture was shaken at 300 r/min for 30 min and filtered through filter paper. The filtrate was then aspirated to induce colorimetric reaction with the TBA, boiled in a water bath (100 °C) for 30 min, and cooled to room temperature (25 °C), whereafter the absorbance values were measured at 532 nm and 600 nm. TBA values were calculated according to Equation (5) [22]:(5)$$\mathrm{TBA}\;(\mathrm{mg}/\mathrm{kg})\;=\frac{A_{532}-A_{600}}{155}\times 72.6\times \frac{1}{10}\times 1000$$
where *A*_532_ is the absorbance at 532 nm and *A*_600_ is the absorbance at 600.

#### 2.13.5. Determination of Total Volatile Basic Nitrogen (TVB-N)

The measurement of the TVB-N was conducted using a method based on a Chinese standard (GB5009.228-2016; Ministry of Agriculture, People’s Republic of China, 2016). Distillation titrations were performed using a fully automatic Kjeldahl nitrogen tester (K1100, Haineng, Shandong, China). The results were expressed as milligrams of nitrogen per 100 g of sample (TVB-N).

#### 2.13.6. Determination of Total Viable Microbial (TVC)

A sample of 1 g of fish was homogenized in 9 mL of sterilized saline to calculate the TVC. Decimal serial dilutions were prepared based on the mixture, and the total active bacteria was tested using the pour plate method, in which 1 mL of the dilution was dropped onto the surface of the plate-counting agar. The plates were then incubated for 24 h at 37 °C. All counts were done using log^10^ CFU/g.

#### 2.13.7. Determination of Endogenous ENZYME Activity

Histone B, histone L and histone D enzyme activity tests were performed according to the instructions of the kits. We followed these steps: First, 2 g of fish samples was homogenized at 12,000 r/min for 30 s, then left for 30 min, and centrifuged at 4 °C for 10 min, after which the supernatant was taken for measurement. Next, standard wells, blank wells and sample wells were set up on the enzyme standard plate, and the corresponding sample solution (50 μL) was added to the standard wells. Then, horseradish peroxidase (HRP)-labeled detection antibody (100 μL) was added to each well, and the enzyme plate was sealed with sealing film and placed in an incubator at 37 °C for 60 min to warm up. The concentrated washing solution was diluted 20 times with distilled water and prepared for use, after which the enzyme plate was removed from the incubator, the liquid was discarded, and the plate was shaken dry. Each well was filled with washing solution and left for 1 min. The liquid was discarded and the well was air-dried, and this process was repeated five times. Thereafter, first chromogen A (50 μL) and then chromogen B (50 μL) were added to each well. They were shaken gently and mixed well, and the color developed at 37 ℃ for 15 min under the avoidance of light. Finally, a termination solution (50 μL) was added to each well to terminate the reaction, as evidenced by the color change from blue to yellow.

The absorbance (optical density, OD) of each well was measured at 450 nm within 15 min, using blank wells for the control and zeroing. The linear regression equation of the standard curve was calculated by the concentration of the standard and the OD value, and the sample concentration was calculated by substituting the OD value of the sample into the equation.

### 2.14. Statistical Analysis

All experiments were repeated in triplicate. Data were processed and analyzed in Excel 2010 and SPSS 24 (SPSS Inc., Chicago, IL, USA) software, respectively, and graphs were created in Origin 8.5. Analysis of variance (ANOVA) and significant differences (Duncan’s multiple comparison procedure) were determinee (*p* < 0.05). These values were expressed as mean ± standard deviation (standard deviation).

## 3. Results

### 3.1. SEM Analysis

The effects of OEO-CSNP addition on the morphology of G/OEO-CSNP film were observed via SEM, and the results are shown in Figure 1. It is obvious from the surface plots that the pure GA film exhibited a smooth, uniform and continuous surface, indicating homogeneity and structural integrity. With increased OEO-CSNP addition, the nanoparticles displayed an agglomeration phenomenon, and the surface of the G/OEO-CSNP composite film developed obvious particles and became rough. At 2%, the G/OEO-CSNP film structure was still complete; it was starting to show particles, although without separation and cracks. At 4%, the G/OEO-CSNP film became rough, and a small number of particles gathered and wrinkles appeared. Finally, at 6–8%, obvious aggregation was observed in the matrices of some of the films, with the surfaces no longer smooth but clearly wrinkled. The aggregation phenomenon observed here is the result of a hydrogen bonding interaction between the chitosan nanoparticles during the drying process [23]. Furthermore, with the increasing addition of OEO-CSNPs, greater contents of OEO were released and its dispersion in the microstructure of thicker films become more inhomogeneous [24]. Hosseini et al. [25] pointed out that in fish gelatin (FG)/CSNP composite films, the roughness of the film surface increases when OEO is added to the matrix. The same was reported by Kumar et al. [26], who found that the addition of nano-silver made the surface of chitosan-gelatin (CS-GL) film rough and the distribution of silver nanoparticles (AgNPs) become non-uniform.

In this study, the pure GA film had no obvious voids at the cross section, however, the addition of OEO-CSNPs caused the GA film structure to become inhomogeneous, with pores of different sizes. The porosity and roughness increased with the increase in OEO-CSNP content, indicating that the nanoparticles were less compatible with GA. The development of cavities may have been due to the formation of non-covalent bonds between the proteins, chitosan and EO polyphenols [27]. Moreover, the gradual increase in the size and quantity of EO volatilized during the drying of the film-forming solution led to slight flocculation and the agglomeration of the substances therein, and the final film was irregular, with an inhomogeneous cross-section [28]. With the increase in OEO-CSNPs, a certain amount of smaller-size nanoparticles penetrated between the different polymers of the films and the voids decreased. This was similar to the findings of Ejaz et al. [29] in which bovine skin gelatin (BSG)/50% clove essential oil (CEO) films were more porous than BSG/25%CEO films, and the addition of zinc oxide nanorods (ZnO NRs) were found to reduce the porosity of both the 25% and 50%CEO BSG films, providing them with excellent structural integrity. Here, the SEM results similarly confirmed that the addition of OEO-CSNPs improved the mechanical properties of the GA films.

### 3.2. XRD Analysis

The crystal structure of the thin film samples as well as the compatibility of gelatin with OEO-loaded CSNPs were investigated using XRD. Figure 2 depicts the XRD patterns of the G/OEO-CSNP films containing OEO-CSNP inclusion complexes. It can be seen that the XRD pattern of the GA film exhibited a sharp main peak at 2θ = 7.6°, the position and intensity of which corresponded to the diameter and content of a three-stranded helical structure [30]. However, the peak did not pan out at different OEO-CSNP additions, and its intensity decreased with increasing OEO-CSNPs, indicating that OEO-CSNPs influenced the three-stranded helix formation. Furthermore, the presence of OEO-CSNPs altered the position and intensity of the second diffraction peak at 2θ = 18°. These results indicate that OEO-CSNPs are able to change the intermolecular structure of GA membranes [31]. The peaks started to become narrower and higher with the increasing nanoparticle concentration, indicating an increase in the crystallinity of the G/OEO-CSNPs films. The XRD patterns of the G/OEO-CSNP films were similar to those of the pure GA films, and no other diffraction spikes appeared, indicating that the OEO-CSNPs were highly compatible with the film-forming matrix GA. These results are consistent with the findings of Almasi [32], which showed that the incorporation of nanoemulsions loaded with marjoram essential oil in pectin films did not significantly change the position of the specific peaks in the XRD patterns of pure pectin. Similarly, Nisar et al. [33] reported that the incorporation of CEO into citrus pectin films did not change the peak positions.

### 3.3. FTIR Analysis

Figure 3 depicts the FTIR spectra of the G/OEO-CSNP films with varying concentrations of OEO-CSNP inclusion complexes. All of the bands observed at 3570–3124 cm^−1^ are associated with N-H and O-H stretching vibrations in the amide-A and phenol functional groups and they indicate the presence of hydrogen bonds [34]. A peak at 2882 cm^−1^ is associated with C-H stretching vibrations of alkane groups in the control GA film (without OEO-CSNPs) [35]. The carbon group stretching of the protein’s amide-I caused the absorption to peak at 1680 cm^−1^.

The peak at 1504 cm^−1^ is attributed to the N-H stretching of amide-II in the GA, while that at 1007 cm^−1^ is attributed to the C-O stretching of primary alcohols [36]. The FTIR spectra of the G/OEO-CSNP composite films were similar to those of the pure GA films. It was observed that some peaks shifted to higher or lower wave numbers as the content of OEO-CSNPs increased. Among them, the absorption peaks of amide-A and amide-II shifted toward higher wave numbers, indicating the possible formation of hydrogen bonds between OEO-CSNPs and the GA matrix. 

In their study on the preparation of chitosan-based nanocomposite films with different concentrations of nanocrystalline cellulose (NCC), Khan et al. [37] suggested that a shift of the amide-A peak toward higher wave numbers was related to the formation of hydrogen bonds between chitosan and NCC. Kanmani and Rhim [35] analyzed the transfer of amide-II peaks to higher wave numbers in GA/clay composite films and reported a strong interaction between the nanoclay and polymer matrices through hydrogen bond formation.

### 3.4. Thickness and Mechanical Property Analysis

The thickness and mechanical properties of the film samples are shown in Table 1. Thickness is an important indicator of the mechanical properties and water vapor permeability of film. In this study, the thickness of the composite film gradually increased from 0.085 mm to 0.092 mm with the increase in OEO-CSNP content. This may have been because the small OEO-CSNPs were uniformly dispersed in the network structure of the GA matrix, filling the gaps between molecules and making the structure of the film more compact. The small particles were also easily agglomerated, leading to an increase in the thickness of the composite film [38].

TS and EB are also important parameters in assessing the magnitude of mechanical properties in film membranes since these are influenced by factors such as drying temperature, the volume of the coating solution, and the interaction between various components [39]. As can be seen from Table 1, the TS of the pure GA was found to be 26.94 ± 0.29 MPa, while the EB was 1.87 ± 0.18%. However, the TS of the G/OEO-CSNPs increased with increasing additions of OEO-CSNPs. This result is consistent with previous reports that the addition of nanoparticles improves the stiffness of GA films and confirms the enhancing effect of nanoparticles on the polymer matrix [40]. Here, the EB also showed an increasing trend with the increase in OEO-CSNPs, rising from 1.87 ± 0.18% to 5.15 ± 0.27%, thus indicating an increase in the flexibility of the GA films. This phenomenon may have been due to the more uniform dispersion of nanoparticles in the GA film matrix, which resulted in a regular arrangement of the GA molecules, and the emergence of a larger specific surface area due to the increase in nanoparticles, which promoted their binding to the GA film. Thus, a low content of OEO-CSNPs could lead to the improvement of the mechanical properties of GA-based films.

### 3.5. Solubility Analysis

As shown in Table 1, the water solubility of the membranes decreased initially, and then increased as the content of OEO-CSNPs increased. The water solubility of OEO-CSNPs decreased by 6.7% compared to pure GA membranes, reaching to 47.99 ± 0.69%, when the water solubility was the lowest. This decrease in water solubility is partly due to the interaction of nanoparticles with the hydrophilic sites of GA, which reduces the number of hydroxyl groups available to absorb water. This result shares similarities with research findings on nanosilver and nanoclay composite GA films. In addition, the decrease in water solubility can also be attributed to the formation of hydrogen bonds between the GA matrix and the nanoparticles, an association that was confirmed by the FTIR results. However, as the content of OEO-CSNPs increased and reached 6%, the bonding between the molecules in the saturated state decreased and the water solubility again increased, a result that was similarly detected in the study by Hosseini et al. [14].

### 3.6. UV-Visible Absorbance and Transmittance Analysis

Visible light and UV light are initiators of oxygen (O_2_) radicals. O_2_ can easily go from the ground state (^3^ O_2_) to the excited state (^1^ O_2_) and react with lipids to form hydroperoxides (ROOH), thereby accelerating the oxidative deterioration of food [41]. The transmittance and absorption values of the films with different amounts of OEO-CSNPs are shown in Figure 4. The UV-vis spectrum of G/OEO-CSNPs is highly similar to that of pure GA films, with optical transparency in the 300–800 nm (low absorbance/high transmittance) range, indicating that GA molecules absorb and scatter to a limited degree in the electromagnetic spectral region. This effect can be attributed to the fact that GA contains fine molecules with cross-linkages (helical regions) much smaller than the wavelength of light [17]. GA becomes optically opaque (high absorbance/low transmittance) in the wavelength range of 200–280 nm. This mainly occurs due to the ability of its protein molecules to block UV and absorb UV-emitting chromophores, especially the aromatic amino acids tyrosine and tryptophan, and to a lesser extent phenylalanine and disulfide bonds [42]. As the concentration of OEO-CSNPs increases, the optical transparency of G/OEO-CSNPs films gradually decreases, and the absorption value shows an increasing trend. However, this phenomenon is dependent on the number of OEO-CSNPs added. A large number of OEO benzene rings can promote the n→π^*^ leap, which sharply decreases the UV transmittance [43]. Furthermore, the aggregation of nanoparticles hinders light transmission. This phenomenon indicates the good UV-visible resistance properties of the G/OEO-CSNPs. Lu et al. [43] found that, as the number of OEO-mesoporous silica nanoparticles (MSNPs) increased, the objects being tested saw their UV-shielding ability increase. Tavassoli attributed the transparencies of nanoparticle-loaded GA films, although slightly smaller than that of GA films, to be very similar to each other and to the NPs in their composition, absorption and scattering of light [17].

### 3.7. Thermal Analysis

The TGA curves of the GA films and the G/OEO-CSNP composite films are shown in Figure 5. Both films exhibited the following three stages of thermal decomposition: The first stage (90 °C) was caused by the evaporation of water; the second, between 190 and 250 °C, was the result of the evaporation of glycerol; while the third stage (320–440 °C) occurred was the most important stage of pyrolysis because of the thermal degradation of the GA matrix at high temperatures [18]. The highest decomposition temperature of the GA films was 354 °C. However, this increased to 420 °C after the addition of OEO-CSNPs. The residual coke contents of the pure GA film with 2%, 4%, 6% and 8% G/OEO-CSNP composite films were 23.8%, 14.47%, 8.15%, 13.11% and 3.34%, respectively, at 600 °C. The final residual content of the G/OEO-CSNP composite film was lower than that of the pure GA film. This was possibly due to the easy volatility of OEO. The amount of residue decreased with the increase in nanoparticle content, suggesting that the addition of OEO-CSNPs enhanced the thermal stability of the GA membranes.

### 3.8. Antioxidation Performance Analysis (DPPH and ABTS Assays)

Free radicals are induced by exogenous factors and their presence leads to the oxidation and deterioration of food. Therefore, active food packaging films should have antioxidant capacity [44]. Carvacrol, the main component of OEO, has been shown to have strong antioxidant activity [45]. The antioxidant properties were assessed on the basis of by DPPH and ABTS free radical scavenging results, as shown in Figure 6. The pure GA films also showed some free radical scavenging activity, which may have been related to the antioxidant properties of GA peptides [18]. The addition of OEO-CSNPs increased the scavenging rate of both DPPH and ABTS radicals in the films. Meng et al. [41] suggested that the high scavenging rate of DPPH radicals and ABTS cationic radicals is related to the good release effect of polyphenols in film materials. The phenolic hydroxyl group of carvallal in OEO can act as a donor of peroxide radicals during oxidation, preventing lipid peroxidation chain reaction and protecting lipids from oxidation [46]. The ABTS radical scavenging ability was reduced when OEO-CSNPs were added at 8%. This could have been because the concentration of OEO in the chitosan nanoparticles was too large, resulting in the coalescence of oil droplets. This enhanced the effect of carvacrol in the OEO and destroyed the structure of the original composite membrane, consequently decreasing the ABTS radical scavenging ability. It was concluded that the addition of OEO-CSNPs to GA films in the preparation of active food packaging films could protect foods from oxidative damage and prolong their shelf life. GA films containing 6% OEO-CSNPs showed better antioxidant properties. Tavassoli confirmed that the addition of quercetin-loaded nanoparticles to gelatin films could solve the problem of oxidation [17].

### 3.9. Antibacterial Activity Analysis

A key property of bioactive packaging materials is their ability to inhibit or prevent the growth of spoilage or pathogenic microorganisms in food. Therefore, in this study, four representative Pseudomonas species were used to evaluate the antimicrobial activity of different GA-based films. The results of these evaluations are shown in Figure 7. Pure GA films showed little or no antimicrobial activity. This was probably due to the capacity of surfactants to denature proteins and disrupt bacterial cell membranes [47]. The G/OEO-CSNPs exhibited antibacterial activity against all four Pseudomonas species, the extent of which depended on the microbial species due to strain specificity [48], as well as on differences in the distribution of OEO-CSNPs in the membrane matrix. Zhang et al. [49] showed that nanoparticles can penetrate bacterial cells, disrupting the cell membrane and intracellular protein synthesis, and can thereby kill the bacterial strain and improve its antimicrobial activity. OEO-CSNPs can release OEO, which can inhibit cell division and kill bacteria, thus exhibiting antimicrobial activity [50]. The G/OEO-CSNPs exhibited some antibacterial activity, even at 2% OEO-CSNP content, and the antibacterial activity of the films increased with increasing OEO-CSNP content. Compared with the preparation of composite films by Hosseini et al. [14] based on the use of chitosan nanoparticles and GA to improve the stability of the composite films, the OEO added in this study conferred excellent antibacterial activity to the GA films, thereby providing a theoretical basis for the development of multifunctional food preservation films.

### 3.10. Effect of Bioactive Film on the Preservation of Mullet

#### 3.10.1. Sensory Evaluation

The changes in sensory scores of different treatment groups on the refrigerated mullet are shown in Figure 8. It can be seen that the sensory qualities of each group continued to decrease as the storage time increased. Initially, the sensory scores of samples from the OEO-CSNP, GA and G/OEO-CSNP groups were slightly lower than those of the CK due to the darker color of the treated fillets. During storage, the sensory scores of the CK group decreased significantly faster than those of the other three groups, showing a greenish color from the sixth day of storage, with a distinct odor of ammonia and a fishy smell on the ninth day. After 12 d of storage, the sensory scores were as follows: G/OEO-CSNPs > OEO-CSNPs > GA > CK. There was a significant difference in odor between the treatment and CK groups by the sixth day of storage, which resulted in a significant decrease in sensory scores. The nanoparticles could play a role in inhibiting microbial growth due to the inclusion of OEO, while the GA membranes could isolate fish from oxygen and delay spoilage. The bio-composite film containing OEO-CSNPs could increase the antiseptic ability of the material, maintain the color of fish, reduce spoilage odor, maintain muscle tissue elasticity, inhibit microbial growth, reduce the rate of spoilage and prolong the shelf life of the oregano fish more effectively than the single OEO-CSNP treatment and GA film.

#### 3.10.2. pH Value

pH is often used as an analytical indicator to detect fish spoilage [51]. In this study, as shown in Figure 9, the trend in pH changes was the same for all groups under different treatments, with a decreasing trend followed by an increasing trend. Initially, the pH decreased due to the glycolytic reactions of glycogen and molecular breakdown of ATP to produce acids [52]. The subsequent increase in pH was attributed to the accumulation of biogenic amines and ammonia produced by microbial and endogenous enzymes, as well as the decomposition of meat proteins during the extended storage time [53]. The initial pH of the fresh fish was 6.57. On day 12, the pH levels of the fish in the CK, OEO-NPs, GA and G/OEO-CSNP groups were 7.03, 6.90, 6.83 and 6.56, respectively. pH changes in the G/OEO-CSNP membrane-treated oregano were slower than in the other groups, indicating that the rate of spoilage of fish treated with the G/OEO-CSNP composite membranes was reduced by the antibacterial properties of the composite membrane, slowing down the production of alkaline substances. Similar trends in pH with increasing storage time were previously reported for grass carp packed with G/OEO-CSNP composite films [54] and carp packed with an alginate coating system and coated with EO emulsions [55].

#### 3.10.3. TBARS Value

The TBARS value is an important indicator of the degree of fat oxidation since it increases with the production of fat oxidation by-products. Figure 9 shows the effect of different treatments on the TBARS of the mullet samples: TBARS values in the different treatment groups showed an increasing trend in the first 6 d of storage; the G/OEO-CSNPs group showed a slow decreasing trend in the middle and late stages of storage; TBARS values of all groups first decreased and then increased; while the G/OEO-CSNPs group increased slowly. A decreasing trend in TBARS values was observed in all groups on day 9, probably due to multiple interaction reactions of malondialdehyde with amino acids, proteins, glucose and other fish components during storage, to produce products that did not react with thiobarbituric acid. Similarly, for example, Jouki et al. [56] found a decrease in TBARS values in rainbow trout after 12 days of refrigeration, after which none of the groups exceeded the limit of 1.0 mg/kg. This study confirmed that the OEO in the G/OEO-CSNPs treatment group could be slowly released into the mullet and delay fat oxidation while providing good antioxidant properties, and that the G/OEO-CSNP composite film could effectively inhibit the fat oxidation process. Other researchers have also demonstrated that films incorporating essential oils have good anti-fat oxidation properties that can delay the release of TBARS. Jouki et al. [56] confirmed that the addition of thyme essential oil or OEO to quince seed mucilage film (QSMF) enhanced its antioxidant properties, and that rainbow trout fillets wrapped with QSMF + thyme or OEO had lower TBARS compared to those fish subject to the effects of the QSMF alone. Chen et al. [57] demonstrated that cinnamaldehyde (β-CD-CI) and thymol (β-CD-Ty) β-cyclodextrin inclusion complexes could be added to the gelatinous base layer of polylactic acid (PLA)/fish gelatin–sodium alginate (FGSA) films, and a lower trend of increasing TBARS values was observed in the bilayer samples containing β-CD-CI/β-CD-Ty compared to those treated with only PLA or PLA/FGSA.

#### 3.10.4. TVB-N

TVB-N is closely related to spoilage microorganisms in fish. Various volatile amines produced during spoilage, such as trimethylamine and ammonia, which are collectively known as TVB-N, are used as quality indicators to evaluate the microbial shelf life of fish [55]. In this study, as shown in Figure 9, the TVB-N values of the different treatment groups showed an increasing trend throughout the storage period. The TVB-N values of the different treatment groups were: G/OEO-CSNPs < OEO-CSNPs < GA < CK. TVB-N levels in the samples packed with G/OEO-CSNPs and OEO-CSNPs films remained below 20 mg/100 g during the storage limit. By day 9, the CK had reached 20.02 mg/100 g, while the TVB-N values of the G/OEO-CSNPs, OEO-CSNPs, and GA groups were 12.06 mg/100 g, 15.90 mg/100 g, and 19.04 mg/100 g, respectively. After 12 d of storage, the TVB-N values of the G/OEO-CSNPs and OEO-CSNPs group were not excessive and the fish TVB-N value of the G/OEO-CSNPs was the smallest after treatment. Since TVB-N is produced mainly by the decomposition of microorganisms present in dead fish, these results were mainly due to the good antimicrobial activity of the composite film. They indicate that G/OEO-CSNPs could effectively inhibit the growth and reproduction of microorganisms, resulting in a slower change in fish TVB-N values, in agreement with the pH change results. Other researchers have similarly demonstrated that blended films, incorporating essential oils with good corrosion resistance, can retard the release of TVB-N. Zhang et al. [58] showed that the TVB-N content of black pork tenderloin, prepared with a blend of curd polysaccharide (CD)/polyvinyl alcohol (PVA)/thyme essential oil films, was significantly lower than that obtained using a pure gel polysaccharide film. Wang et al. [59] reported that STA/PVA films with added OEO-MS were highly effective in maintaining the freshness of sea bass by delaying TVB-N content growth. Thus, this study’s slow-release composite active film showed good potential for aquatic product packaging.

#### 3.10.5. Analysis of Total Bacterial Colony

Microorganisms are the primary cause of aquatic product spoilage and deterioration. As shown in Figure 9, the total number of colonies in all groups increased as the storage time increased. By day 6, the total number of colonies was 6.75 lg CFU/g in the control group and 5.73 lg CFU/g in the gelatin film group. The nanoparticles and composite membrane groups were 5.33 lg CFU/g and 5.27 lg CFU/g, values that did not exceed the acceptable limit for fresh fish. The results showed that the nanoparticles slowed microorganism growth and reproduction in the mullet during storage, thereby reducing the total number of colonies. The nanoparticle composite film had a better antibacterial effect on the fish, working in addition to the antibacterial effect of the nanoparticles themselves, and was able to isolate the external environment and inhibit the growth and reproduction of microorganisms. Furthermore, the sensory analysis results corresponded to the change in total bacterial colony count.

#### 3.10.6. Analysis of Endogenous Enzyme Activity

Endogenous enzymes are key factors in the structural and protein degradation of fish meat, and histoproteinases are an important endogenous enzyme class involved in this process [60]. Zhou et al. [61] reported the strong correlation between histoproteinases and structural changes in fish muscle and found that the various environmental factors to which fish meat is exposed during storage result in multiple endogenous active enzyme changes. Therefore, the combined effects of multiple enzymes were considered in this study when evaluating quality. The activity of histone proteases usually increases over postmortem time and is very active in acidic pH [62]. Figure 10 shows the changes of endogenous protease B, D and L active enzymes with different treatments of oregano. The protease activity of fish flesh treated with CK and GA increased within three days of storage, with peaks of 1.92 and 1.73 times the initial values, respectively. The protease activity of fish flesh treated with NPs and G/OEO-CSNPS increased within nine days of storage, with peaks that are 1.57- and 1.54-fold the initially values, respectively. For histone D, the trend of increase in the four treatment groups remained constant during the first nine days of storage, with peaks of 1.42, 1.36, 1.45 and 1.31 times the initial values for CK, OEO-CSNPs, GA and G/OEO-CSNP treatment groups, respectively. The increasing trends of histone L in the CK, OEO-CSNPs, and GA treatment groups remained constant during the first six days of storage, then underwent a short-term decline and peaked at day 12, with peaks of 1.75, 1.46 and 1.63 times the initial values, respectively. Histone L in the G/OEO-CSNPs treatment group increased to a maximum value of 1.46 times the initial value during the first 6 d.

Overall, histone B, D, and L activities followed the same trend, increasing at the beginning of storage and gradually decreasing at the end of storage, although an increase in histone L was also observed at the end of storage. The magnitude of change in histone B during storage was greater than that of histone D. Similar results were found by Ge [63]: after 15 days of storage, the histone B, B + L, and D activities of grass carp fillets increased in the early stages of storage and gradually decreased at the end of storage. In this study, compared with the control group, the activities of proteases B, D, and L were effectively inhibited in the treated groups. The GA membrane and G/OEO-CSNP composite membrane could inhibit the activities of proteases B, D and L in the frozen fish fillets because the covering effect of the membrane effectively isolated the oxygen in the air, thereby reducing the oxidation of lipids and proteins and decreasing the occurrence of enzymatic activity reactions. Moreover, protease B, D and L activities were strongly inhibited, confirming the strong antioxidant effect of the nanoparticles, which may be related to the active EO ingredients in the nanoparticles. The best preservative effect was observed in the G/OEO-CSNPS treatment group, followed by that of the OEO-CSNPS and, finally, GA groups. Comprehensive analysis showed that the G/OEO-CSNPs could effectively inhibit the elevation of endogenous active enzymes and delay the spoilage of the mullet.

## 4. Conclusions

In this study, G/OEO-CSNPs were prepared by adding various concentrations (0%, 2%, 4%, 6% and 8%) of OEO-CSNPs to GA substrate films, and the optical, physical and biological properties were then tested. The optimal OEO-CSNPs 6% composite membrane, with the best film performance, was subsequently applied to study the preservation of mullet, and its effects on various indexes of the fish samples during cold storage were investigated at 4 °C. The results showed that the OEO-CSNPs were uniformly dispersed in the GA matrix as the nanoparticle content increased. Not only were the film thickness, mechanical capacity, UV/visible light barrier properties and thermal stability significantly enhanced, but they also exhibited excellent antioxidant and antibacterial abilities. When the content of OEO-CSNPs was 6%, the G/OEO-CSNPs properties performed optimally. It was found that when the content of OEO-CSNPs was increased to 6%, G/OEO-CSNPs had the highest DPPH and ABTS radical scavenging rate and the lowest water solubility, and that the antimicrobial activity was not significantly different from that achieved at a rate of 8% OEO-CSNP addition. When the content of OEO-CSNPs increased to 8%, electron microscopy showed that incompatibility occurred between nanoparticles and film-forming molecules. The more essential oil released from nanoparticles, the more dispersed the essential oil was, and the more uneven the film became, resulting in an increasingly rough film surface. Therefore, the comprehensive evaluation selected 6% G/OEO-CSNPs for the application of mullet preservation. In addition, G/OEO-CSNPs with 6% OEO-CSNP content effectively improved the sensory quality of the fish, inhibited the growth and reproduction of bacteria on its surface, slowed the growth of TVC and TVB-N values, delayed lipid oxidation, reduced the TBA value of the fish, and prolonged its shelf life.

In conclusion, in this study, nanoparticles were shown to improve the applicability of GA film as an edible packaging, and had an obvious anti-corrosion effect when applied to the preservation of fish products, thereby indicating its potential to improve the shelf life of packaged food. G/OEO-CSNPs provide green, healthy and safe products for consumers, provide a theoretical basis for quality control and preservation of freshwater fish in the storage process, and open up new ideas for the deep processing of fish products. The study was not sufficiently in depth to investigate the composite film biopreservation mechanism, and subsequent studies should be conducted in terms of flavor compounds, microorganisms, and protein oxidation.

## Figures and Tables

**Figure 1 foods-12-02542-f001:**
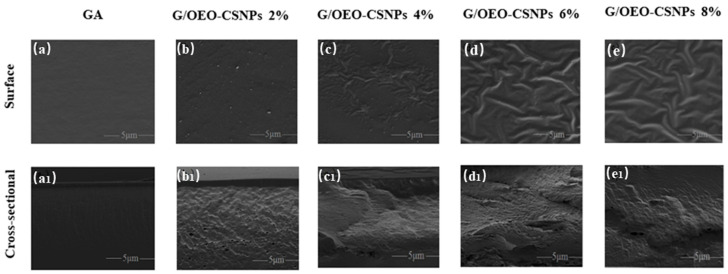
(**a**–**e**) represent 0%, 2%, 4%, 6%, and 8%, respectively surfaces morphologies of the G/OEO-CSNP films. (**a1**–**e1**) represent 0%, 2%, 4%, 6%, and 8%, respectively, across-sectional morphologies of the G/OEO-CSNP films.

**Figure 2 foods-12-02542-f002:**
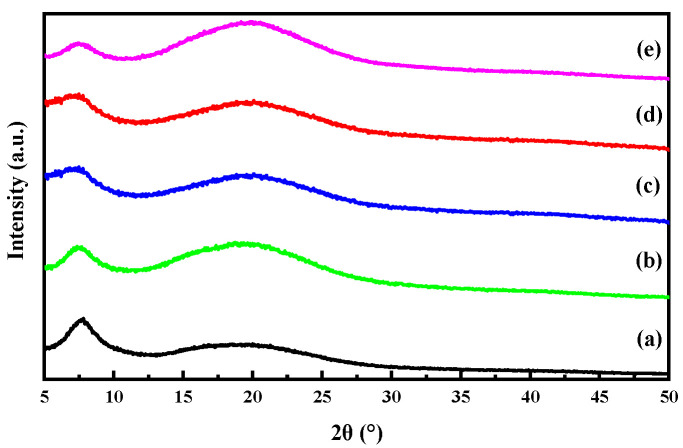
XRD spectra of bioactive films of G/OEO-CSNPs, with (a–e) representing the amount of OEO-CSNPs added as 0%, 2%, 4%, 6% and 8%, respectively.

**Figure 3 foods-12-02542-f003:**
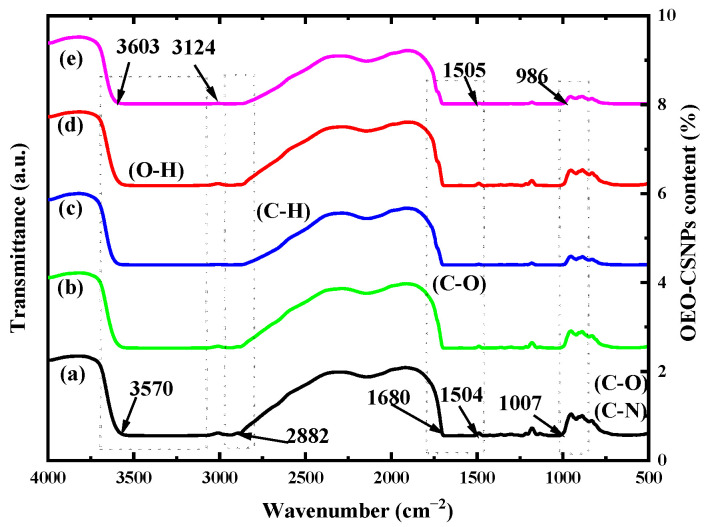
FTIR spectra of bioactive G/OEO-CSNP films, with (a–e) representing the amount of OEO-CSNPs added as 0%, 2%, 4%, 6% and 8%, respectively.

**Figure 4 foods-12-02542-f004:**
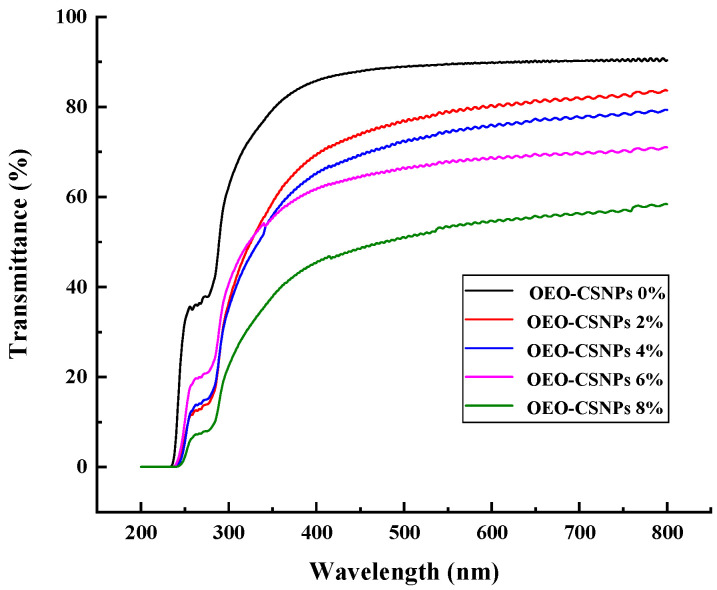
UV-vis light transmittance and absorbance of G/OEO-CSNP bio-composite films.

**Figure 5 foods-12-02542-f005:**
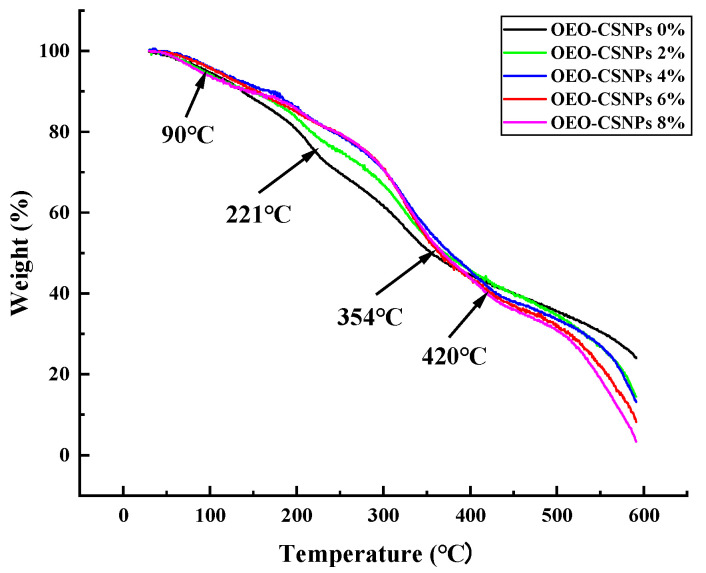
TGA curves of the G/OEO-CSNP bio-composite films.

**Figure 6 foods-12-02542-f006:**
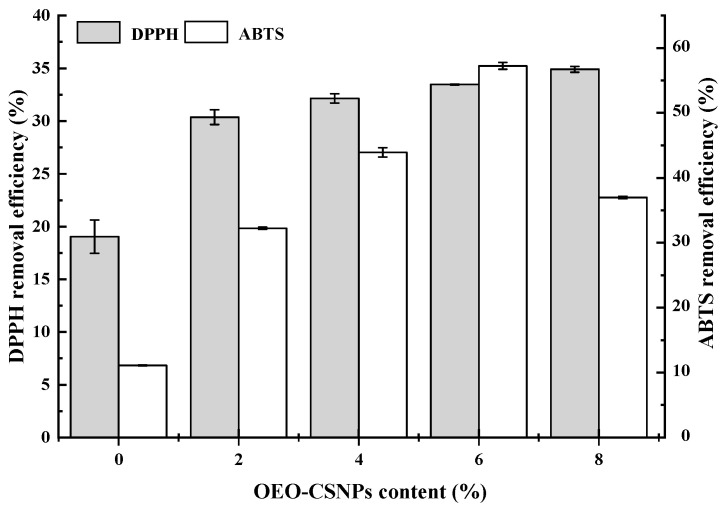
DPPH and ABTS free radical scavenging rates in G/OEO-CSNPs bio-composite films.

**Figure 7 foods-12-02542-f007:**
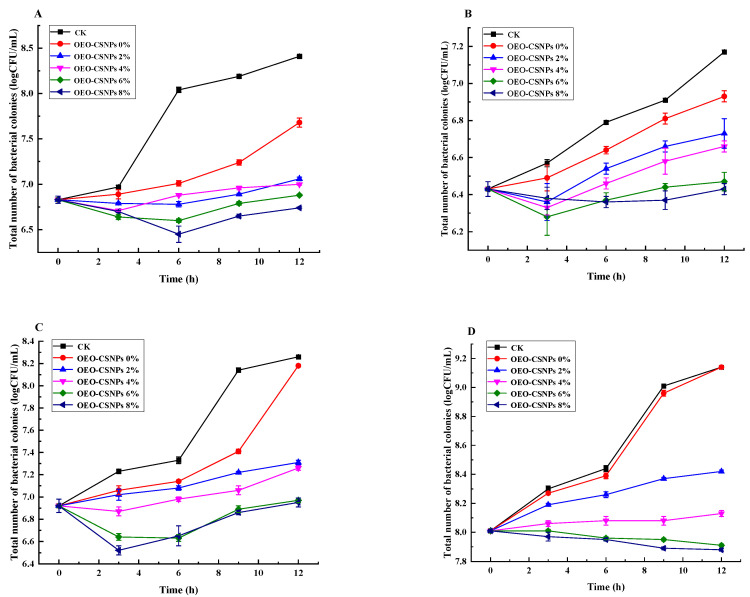
Antibacterial activity of gelatin-based films with different contents of OEO-CSNPs against four Pseudomonas species: (**A**–**D**) represent the dominant spoilage bacteria of oregano. The results are as follows: 11 P. fluorescens, 12 P gessardii, 15 P. fluorescens, and 16 P. jessenii, respectively.

**Figure 8 foods-12-02542-f008:**
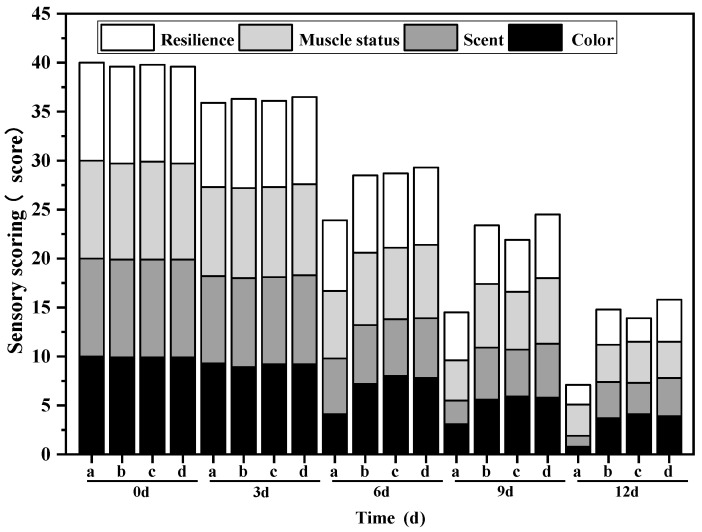
Sensory evaluation of G/OEO-CSNP bio-composite films, with (a–d) representing the CK, OEO-CSNP, GA and G/OEO-CSNP treatment groups, respectively.

**Figure 9 foods-12-02542-f009:**
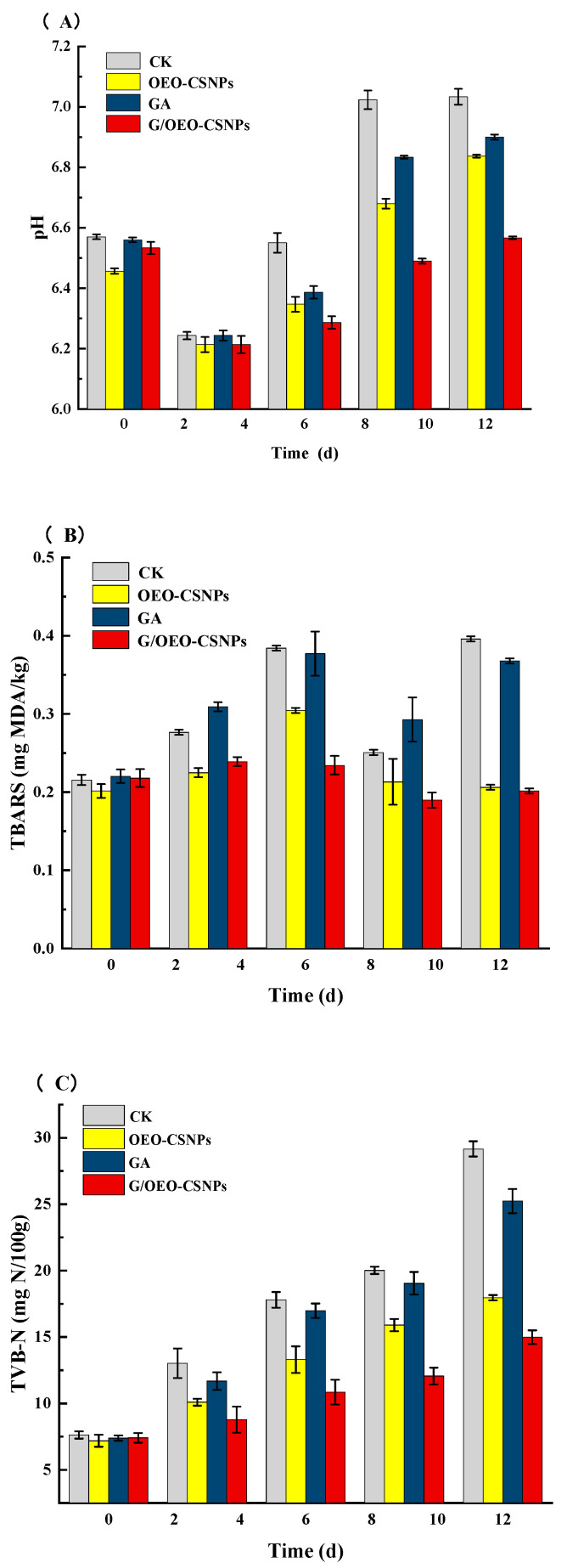
(**A**) pH, (**B**) TBARS, (**C**) TVB-N and (**D**) TVC values in fresh fish samples.

**Figure 10 foods-12-02542-f010:**
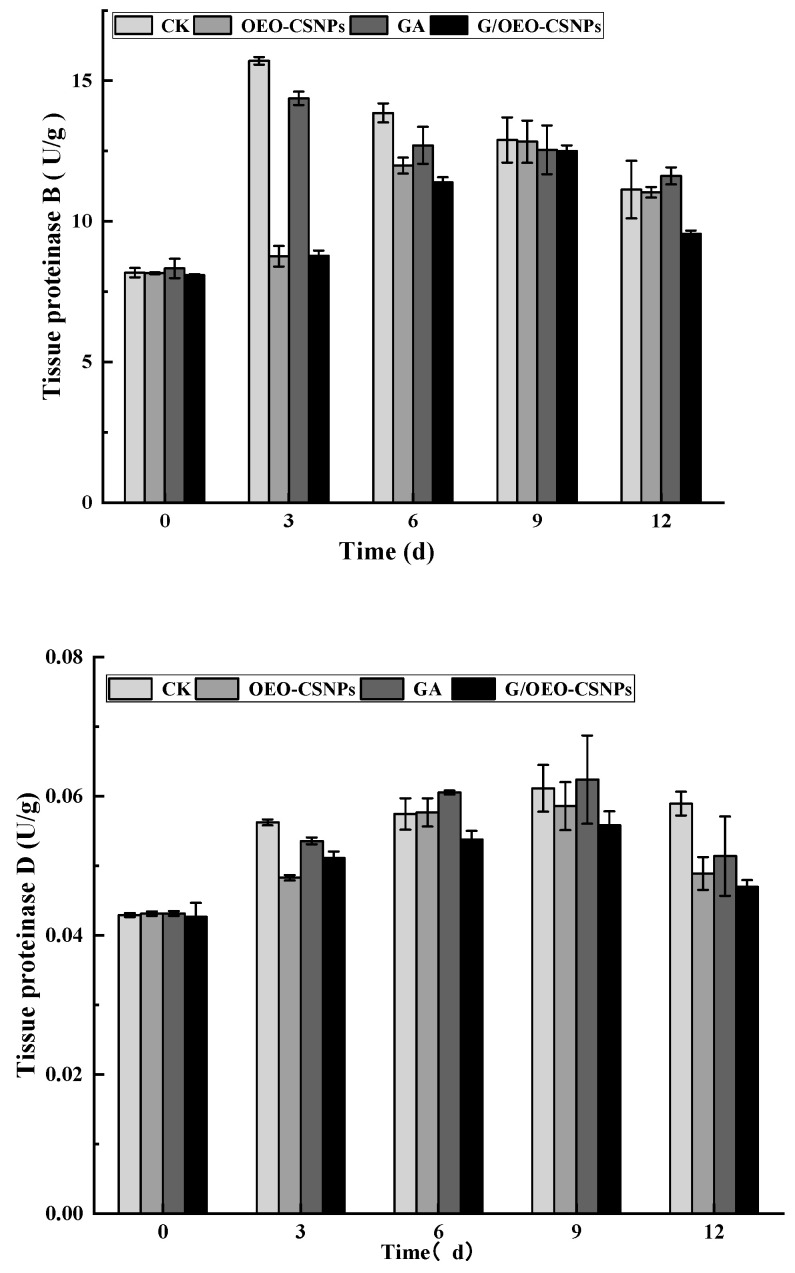
Activity of endogenous proteases (B, D and L) in fresh fish samples.

**Table 1 foods-12-02542-t001:** Physical properties of GA-based films.

OEO-CSNP Content (%)	Thickness (mm)	TS (MPa)	EB (%)	FS (%)
0	0.085 ± 0.001 ^b^	26.94 ± 0.29 ^e^	1.87 ± 0.18 ^e^	54.69 ± 0.77 ^b^
2	0.085 ± 0.002 ^b^	31.83 ± 0.23 ^d^	2.43 ± 0.13 ^d^	54.11 ± 0.44 ^b^
4	0.087 ± 0.003 ^b^	57.31 ± 0.47 ^b^	3.09 ± 0.28 ^c^	49.67 ± 0.69 ^c^
6	0.092 ± 0.001 ^a^	47.62 ± 0.37 ^c^	4.06 ± 0.17 ^b^	48.00 ± 1.11 ^c^
8	0.092 ± 0.003 ^a^	66.14 ± 0.16 ^a^	5.15 ± 0.27 ^a^	57.38 ± 0.91 ^a^

Note: TS (tensile strength), EB (elongation at break), FS (film solubility). Values are expressed as mean ± standard deviation. Lowercase letters in the same column indicate statistically significant differences (*p* < 0.05).

## Data Availability

Data is contained within the article.

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
