# Peer review of "Gelatin Improves the Performance of Oregano Essential Oil Nanoparticle Composite Films—Application to the Preservation of Mullet"

_foods, 2023, doi:10.3390/foods12132542_

Round 1

Reviewer 1 Report

Comments: Good introduction to the work described. It indicates the purpose and scope of the research. The results are interesting and well-explained. However, in some cases, improvements are still needed for the manuscript, according to the following comments.

#1 The authors must provide sufficient detail when describing their materials and methods.

#2 How many parallel measurements were made for each method?

#3 “OEO was purchased from Ji'an Huashen Fragrance Oil Co., Ltd. (Ji'an, China); chitosan (CS) (90% deacetylated) was provided by Shanghai Yuanye Biotechnology Co., Ltd. (Shanghai, China).” These compound characteristics are needed.

#4 In the preparation of the OEO-CSNPs section. More detail about this.

#5 Treatments and concentration determination used in the preparation of bioactive films should be clearly explained. Give the reasons for determining each material’s concentrations used in the study from experimental and relevant reference viewpoints. The authors should explain the reasons for selecting the concentration of OEO-CSNPs (0, 2, 4, 6, and 8%, w/w based on dried GA) used in the study.

#6 It is especially important to justify using some antioxidant activity methods instead of others: a comparison with recent literature would be welcome. (Authors should explain for selecting DPPH and ABTS spectrophotometric assays in the study, instead of other antioxidant activity assays)

#7 Considerations of selecting four dominant bacteria, and excluding other food spoilage bacteria for antimicrobial activity test should be clearly explained.

#8 The authors should explain how to evaluate and determine the optimum formula with the best film characteristics, before applying it to the preservation of mullet.

#9 Compare the obtained findings with similar properties of other bioactive films. More detail about this.

#10 Why is this study important to perform? The authors should highlight the importance of the obtained findings with future trends as applications in the food industry.

Moderate editing of the English language is required.

Reviewer 2 Report

The present manuscript is fairly well written. Nevertheless needs some English revision by a native speaker. On the other hand, some qualifications are noted as average due to some extent is like a zoom into Swarup Roy, Seung-Jae Min, and Jong-Whan Rhim paper of 2023. Reference should be made to them for the study is very similar in objectives, methods and results. No plagiarism is suspected, but a continuation on bio-based functional films with added essential oils they and other authors have been proposing in recent publications on the subject. The quality of the paper is high and rigorous and due merit to the authors is recognized. But some similar papers are being published. Still is a very good paper on an interesting subject. Some similar papers are:

J. Compos. Sci. 2023, 7, 126. https://doi.org/10.3390/jcs7030126

Antibiotics 2022, 11(5), 583; https://doi.org/10.3390/antibiotics11050583

In lines 260-261, the authors mention the inhomogenous pore sizes derived of the inclusion of OE charged nanoparticles and confirmed this observation citing Chu et al. 2020 on line 268. But no discussion is made to wether this observation is a disadvantage or advantage for the application intended. For instance, the antioxidant activity shows a drop in ABTS essay at 8%, ¿could it be due to a nonuniformity of the film? For antioxidant acivity increases in fig 6 for the previous treatments but drops in this level.

Other aspect of interest is the chitosan nanoparticle toxicity. No enzymes are present in humans to degrade chitosan efficiently and nanostructures pose an additional concern. What sizes are present in your film?? . Please revise the following references

Rizeq BR, Younes NN, Rasool K, Nasrallah GK. Synthesis, Bioapplications, and Toxicity Evaluation of Chitosan-Based Nanoparticles. Int J Mol Sci. 2019 Nov 16;20(22):5776. doi: 10.3390/ijms20225776. PMID: 31744157; PMCID: PMC6888098.

Hu YL, Qi W, Han F, Shao JZ, Gao JQ. Toxicity evaluation of biodegradable chitosan nanoparticles using a zebrafish embryo model. Int J Nanomedicine. 2011;6:3351-9. doi: 10.2147/IJN.S25853. Epub 2011 Dec 14. PMID: 22267920; PMCID: PMC3260029.

Extense editing is required. For instance, references are cited by authors´family name but the listing for references are made by number what makes it very difficult to consult. Therefore is mandatory to re-format references listing to avoid a mix of reference system. If the references on text are kept as is, references should be listed in alphabetic order A to Z.

Reviewer 3 Report

In the present study, the authors demonstrated that Gelatin improves the performance of oregano essential oil nanoparticle composite films and his interest to apply it in the preservation of harvested mullet. This is a very interesting contribution for the food and nutrition human, the other hand this work presents some corrections according to the following comments:

Comments to Authors:

1-      The choice of various gelatin (GA) concentrations (0%, 2%, 4%, 6% and 8%) is based on what?

2-      To facilitate the reading and understanding of the paper, in table 1 there are abbreviations that need to be identified in the legend at the bottom of the table.

3-      Figure 4: in the “UV-vis light absorbance” figure, the “absorbance” ordinate axis lacks unity. The same remark for figure 8 the axis of the ordinates misses the unit.

4-      Figure 7 is not cited in the text.

5-      In figure 8 (A), how explains that the pH decreases suddenly between 2-6 days and then suddenly increases between 8-12 days.

6-  In the text the bibliographical references are presented with the names of the authors, on the other hand at the level of the references are classified by numbers. The instructions of the journal should be followed in the presentation of these references.

Reviewer 4 Report

Dear authors

The current study reports an interesting topic on Gelatin improves the performance of oregano essential oil nanoparticle composite films - Application to the preservation of mullet. It shows the need for minor adjustments in its language (standard English). The manuscript's presentation is adequate. Also, all tables mentioned within the manuscript are provided and fit the presented findings and subject.

The Abstract part is clear and well aiming. Minor linguistic adjustments related to a sentence reformulation in a better language are needed. Also, a sentence should be added in the beginning of the abstract body in which authors outline the whole subject under study. On the other hand, all keywords fit. The introduction part is clear, well-structured and aiming. Some adjustments are needed mainly related to a sentence reformulation in a better language besides the need to provide sources (references) for some statements. On the other hand, all the study's aims are easily understood and achieved. The Materials and methods part is well structured and aiming. All needed information and clarifications regarding the adopted methodology are found except the full specification of the used instrumentation. Other minor linguistic adjustments are also requested. The results and discussion part is well structured and aiming. The scientific analysis of the findings is well performed.

Also, correct and adequate statistical approach and presentation were adopted. The discussion of the obtained findings is well aiming and appropriate; authors relied on adequate sources (references) found in literature. Most needed adjustments are related to sentence reformulation in a less cumbersome manner besides other minor linguistic adjustments. The conclusion part is very concise, clear and well aiming. It summarizes appropriately the obtained findings. However, authors should suggest further related researches being based on the raised assumptions from obtained findings.
Briefly, based on the above and below explanations, I find that the manuscript needs minor adjustments and once all following suggestions and recommendations are fully taken into consideration and well addressed.
Abstract
1) The Abstract part is clear and well aiming. Minor linguistic adjustments related to a sentence reformulation in a better language are needed. Also, a sentence should be added in the beginning of the abstract body in which authors outline the whole subject under study. On the other hand, all keywords fit.

2) Kindly add a sentence at the beginning of the abstract body in which you outline the whole subject under study.

3) All keywords fit well.

1. Introduction

1) The Introduction part is clear, well-structured and aiming. Some adjustments are needed mainly related to a sentence reformulation in a better language besides the need to provide sources (references) for some statements. On the other hand, all the study's aims are easily understood and achieved.

2.Materials and methods

1) The Materials and methods part is well structured and aiming. All needed information and clarifications regarding the adopted methodology are found except the full specification of the used instrumentation. Other minor linguistic adjustments are also requested.
2) Kindly provide the full specification (manufacturer and country of origin).

3. Results and discussion

1) The results and discussion part is well structured and aiming. The scientific analysis of the findings is well performed. Also, correct and adequate statistical approach and presentation were adopted. The discussion of the obtained findings is well aiming and appropriate; authors relied on adequate sources (references) found in literature. Most needed adjustments are related to sentence reformulation in a less cumbersome manner besides other minor linguistic adjustments.
It is recommended to cite the following articles:

Control of microbial growth and lipid oxidation in beef using a Lepidium perfoliatum seed mucilage edible coating incorporated with chicory e

4.Conclusion

1) The conclusion part is very concise, clear and well aiming. It summarizes appropriately the obtained findings. However, authors should suggest further related researches being based on the raised assumptions from obtained findings.

Best Regards

-

Reviewer 5 Report

This article with the title “Gelatin improves the performance of oregano essential oil nanoparticle composite films - Application to the preservation of mullet” is useful for food industrial but I would suggest the following:

1-      Abstract must be enriched via valuable results which pave the way for understanding the audiences

2-      Line 75; explain more about preparation of OEO-CSNPs

3-      Line 129; You have not introduced M2

4-      Line 134; explain more about method of Tavassoli et al (2021) (how to measure UV-visible absorbance and transmittance).

5-      Line 338; Abbreviation TS and EB is given, because you are mentioning these words for the first time, bring them in full format

6-      Conclusion is very short and lack the basic fundamentals of the results obtained. Please, authors should re-write the conclusions again with more emphasis on the significant comparison and the improvements from the results obtained.  

-

Round 2

Reviewer 3 Report

The authors have made all the necessary corrections and have also answered all the questions.